# Overcoming mutation-based resistance to antiandrogens with rational drug design

**Minna D Balbas[1,2], Michael J Evans[2], David J Hosfield[3], John Wongvipat[2], Vivek K Arora[2], Philip A Watson[2], Yu Chen[2], Geoffrey L Greene[3], Yang Shen[4]\*, Charles L Sawyers[2,5]\***

[1]Louis V. Gerstner, Jr. Graduate School of Biomedical Sciences, Memorial Sloan-Kettering Cancer Center, New York, United States; [2]Human Oncology and Pathogenesis Program, Memorial Sloan-Kettering Cancer Center, New York, United States; [3]Ben May Department for Cancer Research, University of Chicago, Chicago, United States; [4]Toyota Technological Institute at Chicago, Chicago, United States; [5]Howard Hughes Medical Institute, Memorial Sloan-Kettering Cancer Center, New York, United States

**Abstract** The second-generation antiandrogen enzalutamide was recently approved for patients with castration-resistant prostate cancer. Despite its success, the duration of response is often limited. For previous antiandrogens, one mechanism of resistance is mutation of the androgen receptor (AR). To prospectively identify AR mutations that might confer resistance to enzalutamide, we performed a reporter-based mutagenesis screen and identified a novel mutation, F876L, which converted enzalutamide into an AR agonist. Ectopic expression of AR F876L rescued the growth inhibition of enzalutamide treatment. Molecular dynamics simulations performed on antiandrogen–AR complexes suggested a mechanism by which the F876L substitution alleviates antagonism through repositioning of the coactivator recruiting helix 12. This model then provided the rationale for a focused chemical screen which, based on existing antiandrogen scaffolds, identified three novel compounds that effectively antagonized AR F876L (and AR WT) to suppress the growth of prostate cancer cells resistant to enzalutamide.

**\*For correspondence:**
yangshen@ttic.edu (YS);
sawyersc@mskcc.org (CLS)

**Reviewing editor**: Christopher Glass, University of California, San Diego, United States

## Introduction

The recent FDA approval of enzalutamide (formerly MDV3100) confirms the continued critical role AR signaling plays in castration-resistant prostate cancer (*Tran et al., 2009*; *Scher et al., 2012*). In spite of these promising results, patient responses to enzalutamide are varied and often short lived. Reactivation of AR signaling has been implicated in resistance to previous antiandrogen therapy (*Linja et al., 2001*; *Chen et al., 2004*), and one well-documented mechanism of reactivation is point mutation in the ligand-binding domain (LBD) of AR (*Bergerat and Ceraline, 2009*). Many of these mutations broaden ligand specificity, and some confer resistance by converting the AR antagonist into an agonist of the mutant receptor (*Veldscholte et al., 1990*; *Haapala et al., 2001*; *Hara et al., 2003*). Because previous work with targeted therapies that inhibit oncogenic kinases has shown that unbiased mutagenesis screens in preclinical models can identify a priori clinically relevant mutations that alter drug activity (*Azam et al., 2003*; *Burgess et al., 2005*), we designed a novel screening method to prospectively identify AR mutations that confer resistance to enzalutamide.

Mutagenesis screens to identify kinase inhibitor-resistant alleles of kinase targets such as BCR-ABL have relied upon cytokine-dependent test cells that become cytokine independent after introduction of the target kinase. Cells expressing drug-resistant kinase alleles selectively expand in the presence of drug, allowing rapid identification of mutations that confer drug resistance. There is no comparable

**eLife digest** Prostate cancer is the most commonly diagnosed cancer in men, and the second most lethal. All stages of prostate cancer depend upon male sex hormones, also known as androgens, to grow because these hormones bind and activate androgen receptors. A class of drugs termed 'antiandrogens' can effectively treat prostate cancer because they bind to androgen receptors without activating them, thereby preventing androgens from binding. However, the efficacy of even highly potent antiandrogen drugs, such as enzalutamide is short-lived in many patients, and understanding the biological mechanisms that cause drug resistance is one of the major objectives in translational prostate cancer research.

Resistance can arise through mutations of the androgen receptor that result in the receptor being activated, rather than inhibited, by antiandrogen drugs. However, no such mutations are known yet for enzalutamide, and researchers are keen to understand whether they exist and, if so, to generate new drugs for prostate cancer that overcome them. To identify mutations that may lead to resistance, Balbas et al. designed a new screening method in human prostate cancer cells and showed that androgen receptors with a specific mutation (called F876L) can be activated by enzalutamide. More comprehensive biological studies showed that prostate cancer cells harboring the mutation continued to grow when treated with the drug. Balbas et al. also showed that this mutation can arise spontaneously in human prostate cancer cells treated long term with enzalutamide.

Balbas et al. reasoned that the mutation likely altered the way enzalutamide binds to the androgen receptor, and used computer-guided structural modeling of the complex formed by the receptor and the drug to investigate how this might occur. These studies indicated that the region of the androgen receptor containing the F876L mutation comes into direct contact with the drug, and provided a structural explanation for the loss of inhibition. Because these studies showed how enzalutamide might bind to the androgen receptor, they also suggested ways in which enzalutamide could be chemically modified to restore its inhibitory activity against the mutant receptor.

Balbas et al. then designed and synthesized a set of novel compounds, which the modeling data suggested could act as inhibitors of the mutant receptor. Several of these compounds inhibited the activity of both mutant and wild-type forms of the androgen receptor, and suppressed the growth of both enzalutamide-resistant and nonresistant prostate cancer cells.

The work of Balbas et al. outlines a general screening strategy for the discovery of clinically relevant mutations in cancer genes, and shows how in silico technologies can accelerate drug discovery in the absence of a crystal structure of a protein–drug complex. It also emphasizes how understanding the manner in which a drug binds its target can stimulate rational design of improved drug candidates.

strategy available for antiandrogens because introduction of AR does not confer a comparable growth advantage in AR-negative cells. We instead chose to identify and select cell populations with persistent AR transcriptional activity in the presence of enzalutamide. We reasoned that targeting the biological process of interest (transcriptional activation of a target gene), rather than a distal symptom of resistance to drug (i.e., persistent viability, elevated proliferation) might identify resistant clones more quickly. To this end, an AR-regulated EGFP reporter, with a probasin promoter and PSA enhancer elements driving EGFP expression (Pb.PSE.EGFP) (*Chapel-Fernandes et al., 2006*), was used to screen for and enrich cell populations bearing biologically active mutations (*Figure 1A*).

## Results

Proof-of-concept studies were conducted with LNCaP cells cotransduced with Pb.PSE.EGFP and AR W741C, a well-characterized mutation that converts the AR antagonist bicalutamide into an agonist (*Haapala et al., 2001*; *Hara et al., 2003*). With starting ratios of 1:100 and 1:1000 cells overexpressing AR W741C to wild-type AR, respectively, we sorted cells with FACS that maintained EGFP expression after acute bicalutamide exposure. After four rounds of bicalutamide 'selection' and sorting, LNCaP cells expressing AR W741C dominated the final populations (*Figure 1—figure supplement 1*).

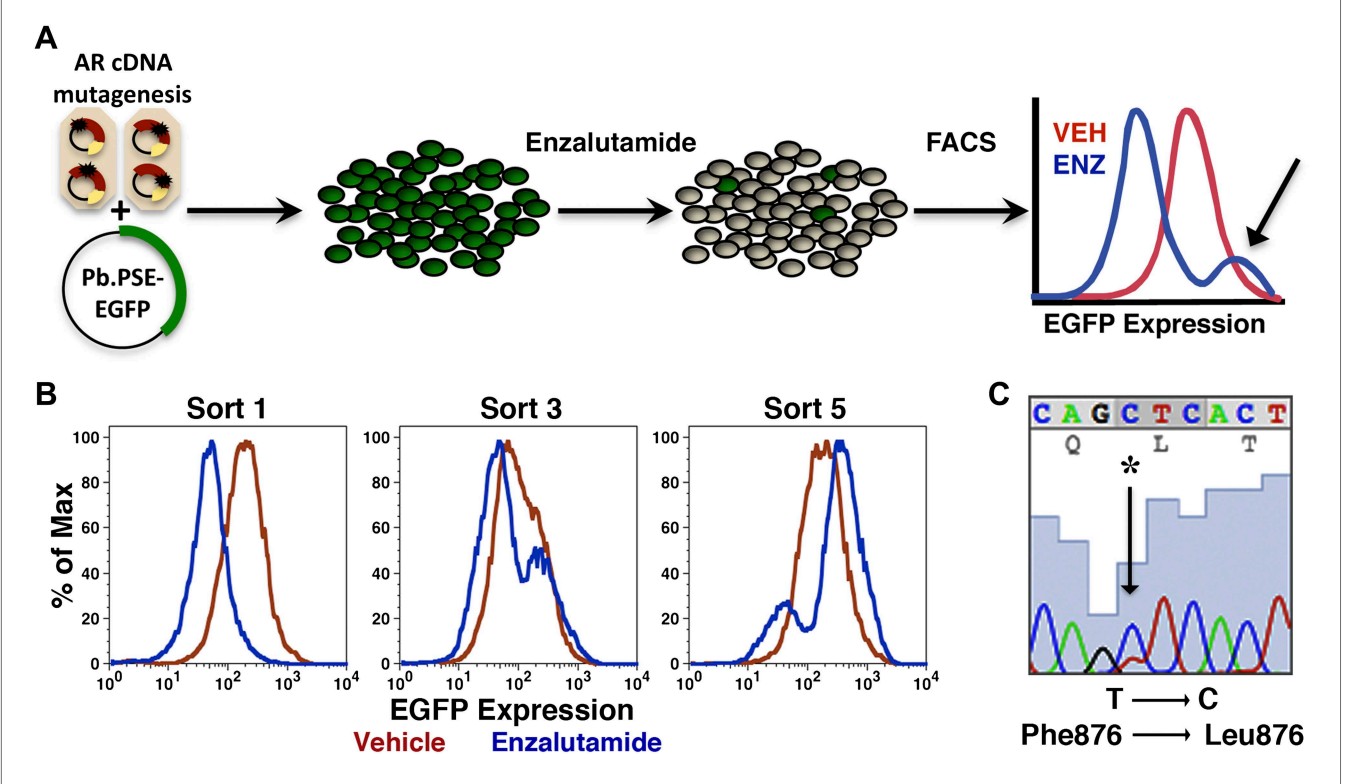

**Figure 1**. Mutagenesis screen for enzalutamide resistance identifies novel AR mutation. (**A**) A cartoon of the AR mutagenesis screen developed to identify enzalutamide resistance mutations. Briefly, cells were cotransduced with a randomly mutagenized AR cDNA library (AR*) and EGFP reporter of AR activity (Pb.PSE.EGFP), treated with 1 µM enzalutamide, and EGFP-positive cells were sorted using FACS. AR was PCR amplified and sequenced to identify relevant mutations. (**B**) Representative FACS histograms showing the progressive enrichment of an EGFP-positive subpopulation of LNCaP/AR*/Pb.PSE.EGFP cells post multiple rounds of enzalutamide treatment and cell sorting. (**C**) A Sanger sequencing trace of exon 8 within AR on the exogenous AR allele from LNCaP/AR*/Pb.PSE.EGFP cells after the fifth FACS sort. The position of the mutation is highlighted with an arrow. The alignment was performed against AR WT. AR: androgen receptor.

The following figure supplements are available for figure 1:

**Figure supplement 1**. Enrichment of AR W741C mutant expressing LNCaP-Pb.PSE.EGFP cells after bicalutamide treatment and EGFP sorting.

**Figure supplement 2**. Endogenous AR target gene, PSA is induced by enzalutamide in FACS-sorted cells.

**Figure supplement 3**. Expression of EGFP and endogenous AR target gene FKBP5 remain AR-dependent in FACS-sorted cells.

**Figure supplement 4**. AR F876L mutation accounts for 50% of AR in cells after fourth sort, and further enriched after the fifth sort.

We next conducted the enzalutamide resistance screen with the Pb.PSE.EGFP reporter and a randomly mutagenized AR library. After five iterations of enzalutamide exposure and FACS sorting, we identified a population of cells with durable EGFP expression (***Figure 1B***). Moreover, enzalutamide promoted AR transcriptional activity in these cells, reflected by induction of EGFP expression compared to vehicle control (***Figure 1B***). Analysis of endogenous AR target gene expression confirmed that enzalutamide behaved as an agonist in the enriched cell population (***Figure 1—figure supplement 2***), and siRNA knockdown of AR showed that these pharmacologically induced changes remained AR dependent (***Figure 1—figure supplement 3A,B***).

To identify AR mutations in these cells, we amplified the exogenously expressed AR cDNA, and Sanger sequenced the PCR product. In two of three replicates, a single dominant point mutation emerged, resulting in the amino acid substitution F876L (***Figure 1C***). Importantly, this mutation clearly enriched throughout the selection process (***Figure 1—figure supplement 4***).

To validate the results of the screen, an AR F876L vector was engineered and transduced into parental LNCaP cells expressing the Pb.PSE.EGFP reporter. Treatment of these cells with enzalutamide resulted in a dose-dependent induction of EGFP expression (*Figure 2A*, *Figure 2—figure supplement 1*). We also introduced AR F876L cDNA into AR-negative CV1 cells along with an AR-dependent luciferase construct, and upon enzalutamide treatment, luciferase activity was induced ~50-fold (*Figure 2B*). These results were comparable to those seen with the previously reported AR mutations T877A and W741C, which confer agonism to hydroxyflutamide and bicalutamide, respectively. Moreover, enzalutamide treatment potently induced nuclear localization of AR F876L (*Figure 2—figure supplement 2A*), and chromatin immunoprecipitation studies showed that enzalutamide recruited AR F876L to the enhancers of AR target genes (*Figure 2—figure supplement 2B*). Consistent with these results, endogenous AR target gene expression was either no longer repressed by enzalutamide (*Figure 2C*, left) or strongly induced by enzalutamide in cells expressing AR F876L (*Figure 2C*, right). A competition assay with 16β[$^{18}$F]fluoro-5α-DHT (18F-FDHT), to measure relative AR binding affinity (*Tran et al., 2009*), showed that enzalutamide binds with higher affinity to AR F876L than wild-type AR (*Figure 2—figure supplement 3*), similar to what has been shown for hydroxyflutamide and the AR T877A mutant (*Ozers et al., 2007*). Notably, F876L similarly impacted the pharmacology of ARN-509 (*Clegg et al., 2012*), a structurally discrete antiandrogen sharing the bisaryl-thiohydantoin core motif (*Figure 2B*, *Figure 2—figure supplements 4, 5*).

In vitro growth assays were conducted to examine the consequences of AR F876L expression on enzalutamide sensitivity in prostate cancer cell lines. Although enzalutamide treatment potently inhibits the growth of parental VCaP cells (*Tran et al., 2009*), overexpression of AR F876L entirely reversed this phenotype (*Figure 2D*). Enzalutamide also rescued the growth of VCaP/AR F876L cells in androgen-depleted media, similar to that seen with the endogenous androgen DHT (*Figure 2E*). Finally, these results were recapitulated in CWR22Pc cells, another prostate cancer cell line that is sensitive to enzalutamide (*Figure 2—figure supplement 6A,B*).

In vivo, LNCaP/AR cells overexpressing either AR WT or F876L were grafted subcutaneously into castrate SCID mice and time to tumor emergence/progression was determined in the presence or absence of drug. While the growth of wild-type AR tumors was almost completely inhibited by enzalutamide treatment, tumors expressing AR F876L grew rapidly in the presence of enzalutamide, similar to vehicle-treated tumors of either genotype (*Figure 2F*).

We next asked whether the F876L mutation spontaneously arises in antiandrogen-sensitive human prostate cancer models after prolonged treatment with enzalutamide or ARN-509. After culturing CWR22Pc cells in vitro with enzalutamide for several months, more than 50% of the cells expressed the F876L mutation (*Table 1*). Prolonged culture of these cells with ARN-509 also selected for a small population (~1.3%) expressing AR F876L. In vivo, long-term enzalutamide or ARN-509 therapy in mice bearing LNCaP/AR xenograft tumors also resulted in the outgrowth of tumor cell populations expressing AR F876L (*Table 1*). Sequencing revealed that AR F876L predominated in one tumor (~71%), was present at low frequency in four other tumors (~1 to 2%), and that a distinct amino acid substitution at this residue, F876I, was enriched in one enzalutamide-resistant tumor (*Table 2*).

To further explore the function of Phe876 as the 'gateway' residue governing enzalutamide and ARN-509 pharmacology, we used site-directed mutagenesis to make additional amino acid substitutions at residue 876. A conservative F876Y substitution did not alter the pharmacology of either drug, but aliphatic substitutions structurally similar to F876L, such as F876I (also found in one xenograft with acquired resistance), conferred agonism to both enzalutamide and ARN-509 (*Figure 2—figure supplement 7*).

Notably, we observed that bicalutamide did not induce AR F876L transcriptional activity in our luciferase reporter assay, either at low (*Figure 2B*) or at high (*Figure 2—figure supplement 7*) concentrations, suggesting that it retains weak antagonist activity against this mutant. We conducted EGFP reporter assays to determine if AR F876L transcriptional activity is inhibited by bicalutamide, and found that while at low doses, it is an effective inhibitor of AR F876L, it loses potency at higher concentrations (*Figure 2—figure supplement 8*). We also observed only minimal growth inhibition in CWR22Pc cells expressing AR F876L with bicalutamide treatment (*Figure 2—figure supplement 6A*). These data, along with the knowledge that AR overexpression is a common resistance mechanism in patients with CRPC (*Linja et al., 2001*) and confers partial agonism on bicalutamide (*Chen et al., 2004*), suggest that bicalutamide is not a viable treatment option for patients who fail on enzalutamide due to AR F876L mutation.

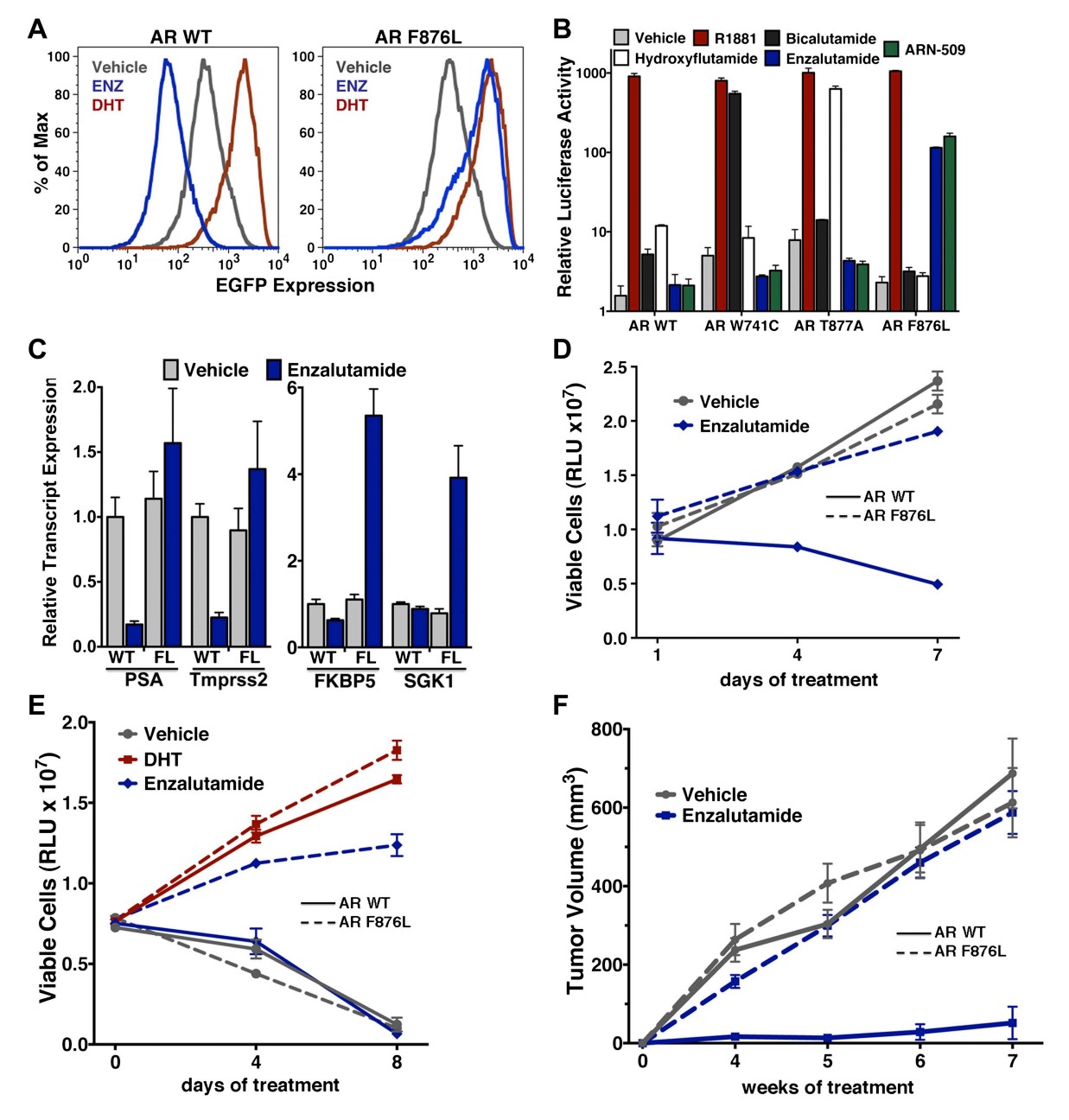

**Figure 2**. AR F876L mutation converts enzalutamide into an agonist and rescues enzalutamide-induced growth inhibition. (**A**) A representative FACS histogram shows the induction of AR-dependent EGFP expression by enzalutamide in LNCaP-Pb.PSE.EGFP cells ectopically expressing AR F876L. The magnitude of induction by enzalutamide (10 μM) is comparable to that conferred by the endogenous androgen DHT (1 nM). Enzalutamide treatment of LNCaP-Pb.PSE.EGFP cells ectopically expressing AR WT effectively suppressed EGFP expression. Geometric-mean fluorescence intensity for WT treated cells: vehicle (348), enzalutamide (66.4), DHT (1554); for F876L cells: vehicle (345), enzalutamide (1051), DHT (1699). (**B**) Cotransduction of CV1 cells with an AR-regulated firefly luciferase construct, a constitutive Renilla luciferase construct, and one of the indicated AR constructs, recapitulates the pharmacology observed in the EGFP reporter system. These cells were treated with vehicle (DMSO), antiandrogens (1 μM), or the synthetic androgen R1881 (1 nM). A dual luciferase assay was conducted on cell lysates, the firefly signal was normalized to the constitutive Renilla activity, and the data are reported as relative light units (RLUs). Notably, the bisaryl-thiohydantoin antiandrogens (enzalutamide and ARN-509) effectively induce AR F876L transcriptional activity, while structurally discrete antiandrogens (hydroxyflutamide and bicalutamide) do not impact AR F876L activity in this assay. As expected, the transcriptional activity of AR W741C or AR T877A was induced by bicalutamide or hydroxyflutamide, respectively. (**C**) Quantitative reverse transcription–polymerase chain reaction analysis of LNCaP/AR F876L cells shows that enzalutamide (1 μM) can induce the expression of canonical AR-regulated gene products (i.e., *PSA*, *TMPRSS2*, *SGK1*, and *FKBP5*). Relative gene expression post therapy for LNCaP/AR WT cells is included as

*Figure 2. Continued on next page*

*Figure 2. Continued*

positive controls. FL = AR F876L, data are normalized to GAPDH and represented as mean ± SD, *n* = 3. (**D**) Cell proliferation data shows that overexpression of AR F876L in a human prostate cancer cell line sensitive to enzalutamide therapy can rescue cell growth. VCaP cells overexpressing either AR WT (solid lines) or AR F876L (dashed lines) were cultured in media containing full serum, treated with either vehicle (DMSO) or 10 μM enzalutamide, and the viable cell fraction was determined at the indicated time points (data is represented as mean ± SD, *n* = 3). (**E**) Cellular proliferation data shows that enzalutamide also rescues the growth of VCaP cells expressing AR F876L in androgen-depleted media. VCaP cells overexpressing either AR WT (solid lines) or AR F876L (dashed lines) were treated with vehicle (DMSO), 1 nM DHT, or 10 μM enzalutamide, and the viable cell fraction was determined at the indicated time points (mean ± SD, *n* = 3). (**F**) A time to progression study for mice bearing subcutaneous LNCaP/AR-WT (solid lines) or LNCaP/AR-F876L (dashed lines) xenografts further highlights the genotype-dependent pharmacology of enzalutamide. Inoculated animals were treated once daily through oral gavage with either vehicle or enzalutamide (30 mg/kg), and tumor size was monitored weekly (11–16 tumors per treatment group). While enzalutamide potently suppressed the growth of LNCaP/AR-WT tumors, LNCaP/AR-F876L tumors exposed to enzalutamide grew with kinetics roughly equivalent to either vehicle treatment arm. AR: androgen receptor; WT: wild-type.

The following figure supplements are available for figure 2:

**Figure supplement 1**. Dose-dependent induction of EGFP expression by enzalutamide in LNCaP-Pb.PSE.EGFP cells expressing AR F876L.

**Figure supplement 2**. Enzalutamide induces AR F876L nuclear translocation and DNA binding to AR enhancer elements.

**Figure supplement 3**. Enzalutamide binds to AR F876L with higher affinity than to AR WT.

**Figure supplement 4**. AR F876L mutation converts ARN-509 into an AR agonist.

**Figure supplement 5**. Dose-dependent agonism in AR F876L expressing cells treated with enzalutamide or ARN-509.

**Figure supplement 6**. Ectopic expression of AR F876L in CWR22Pc cells confers resistance to enzalutamide and rescues growth in androgen-depleted media.

**Figure supplement 7**. Other amino acid substitutions at Phe876 modify the pharmacology of second-generation antiandrogens.

**Figure supplement 8**. Bicalutamide is a modest inhibitor of AR F876L transcriptional activity.

That F876L so dramatically impacted the pharmacology of enzalutamide and ARN-509 suggested that a clear structural change in the drug–receptor complex might be occurring. This consideration prompted us to investigate the structural basis of this antagonism-to-agonism conversion. Because a crystal structure depicting AR bound to an antagonist does not yet exist, we performed structural modeling using ligand docking and molecular dynamics (MD) simulations (***Karplus and McCammon, 2002***; ***Jorgensen, 2004***). In designing the study, we noted that both enzalutamide and ARN-509 share identical A-rings with bicalutamide (***Figure 3A***) and its derivative S1 that were respectively cocrystallized with the LBD of AR W741L and AR WT in agonist conformations (PDB ID 1Z95 and 2AXA) (***Bohl et al., 2005a***, ***2005b***). 2AXA was chosen as a structural template as it bears fewer amino acid substitutions compared to 1Z95. After initial quantum-mechanical geometry optimization of the small molecules, each was independently docked into AR WT or AR F876L with the mutually shared A-ring overlaid with that of S1, whereupon 10-ns explicit-solvent MD simulations were performed.

The docked enzalutamide and ARN-509 molecules demonstrated strikingly different interaction patterns with AR compared to bicalutamide (***Figure 3B,C***). Notably, the thiohydantoin B-ring prevents the compound from accessing the 'H12 pocket' occupied by bicalutamide. Instead, the conformationally restricted thiohydantoin forces the C-ring to bind a region near the C terminus of helix 11 and the loop connecting helices 11 and 12, that we termed the 'H11 pocket'.

**Table 1.** Enrichment of F876 mutant expressing cells after enzalutamide or ARN-509 exposure

| | F876 mutation* | Treatment |
|---|---|---|
| CWR22PC (in vitro) | 52% of all reads | Enzalutamide |
| | 1.3% of all reads | ARN-509 |
| | not detected | Vehicle |
| LNCaP/AR (in vivo) | 3/8 tumors* | Enzalutamide |
| | 3/14 tumors | ARN-509 |
| | 0/5 tumors | Vehicle |

*One tumor F876I, all others F876L

**Table 2.** F876 mutation frequency in drug-resistant LNCaP/AR xenograft tumors

| Treatment | Frequency (%) | Mutation | Endogenous or exogenous locus |
|---|---|---|---|
| Enzalutamide | 1.06 | F876L | Exogenous |
| Enzalutamide | 1.82 | F876L | Exogenous |
| Enzalutamide | 2.19 | F876I | Exogenous |
| ARN-509 | 1.10 | F876L | Exogenous |
| ARN-509 | 1.39 | F876L | Exogenous |
| ARN-509 | 71.23 | F876L | Endogenous |

As seen in the MD simulations using the WT receptor with enzalutamide and ARN-509 (*Figure 3D,E* [in red], *Figure 3—figure supplement 1*), accommodation of the C-ring in this region is coupled to significant conformational rearrangements of residues on H11 and the H11–H12 connecting loop that prevents H12 from adopting the agonist conformation required for efficient coactivator recruitment. To investigate how the F876L mutation might alleviate antagonism, we performed similar MD simulations using the F876L receptor (*Figure 3D,E* [in cyan], *Figure 3—figure supplement 1*). For WT receptors in complex with enzalutamide and ARN-509, the average RMSDs to the crystal agonist conformation for the helix 11 terminus (residues 875–882) were measured at 2.24 and 1.94 Å, respectively, and those for the helix 12 terminus (residues 893–900) were 1.81 and 2.08 Å, respectively. For the F876L mutants, in comparison, the average RMSDs for helix 11 were somewhat lower at 1.01 and 1.70 Å for enzalutamide and ARN-509, respectively, and those for helix 12 went down to 1.37 Å for both ligands. The results demonstrate that despite inducing similar dislocations in the H11 pocket, the mutation allows the receptor to reposition H12 in a more agonist-like conformation that is compatible with coactivator recruitment (*Figure 3D,E*, *Figure 3—figure supplement 1*). A close look in the H11 pocket (*Figure 3—figure supplement 2A,B*) indicated the following: (1) F876 in AR WT likely interacts with the C-ring end of enzalutamide or ARN-509 through favorable pi stacking or van der Waals contacts, and (2) the loss of such favorable contacts upon F876L mutation concurrently affects the conformational choices of helices 11 and 12, which interact through both bonded and nonbonded forces. Importantly, these structural modeling results are consistent with the differential resistance profiles for enzalutamide and bicalutamide involving residues 741 and 876, respectively (*Figure 2B*).

Another notable insight from these simulations was that the substituent on position 4 of the B-ring (i.e., the geminal [gem]-dimethyl group on enzalutamide, the spirocyclobutyl ring on ARN-509; *Figure 4A*) was predicted to lie in close proximity to residues on H12 of the mutant receptor (*Figure 3D,E*). Moreover, the steric girth of the substituent appeared to impact the positioning of H12, as the bulkier spirocyclobutyl moiety on ARN-509 elicited greater H12 displacements in AR WT than did enzalutamide's gem-dimethyl group (*Figure 3E*).

To restore the positioning of H12 into an antagonist conformation for AR F876L, we designed and synthesized a series of analogues bearing saturated hydrocarbon spirocycles of incrementally greater size and complexity on the B-ring of the enzalutamide scaffold (DR100-103, *Figure 4A*). We defined these ring extensions on the B-ring as the D-ring (*Figure 3A*). The merit of this approach was also supported by prior medicinal chemistry, which had shown that discrete bisaryl-thiohydantoin compounds bearing similar D-rings, were effective antagonists of AR WT (*Jung et al., 2010*). In this regard, we were optimistic that, minimally, the larger inhibitors we designed should be tolerated within the ligand-binding pocket.

Consistent with this precedent, the DR100-103 series inhibited the transcriptional activity of AR WT in the EGFP reporter assay (*Figure 4A*, *Figure 4—figure supplement 1A,B*). Whereas DR100-102 behaved as strong agonists for AR F876L (*Figure 4—figure supplement 1A*), (±)-DR103 potently inhibited the mutant receptor and antagonized DHT induction (*Figure 4—figure supplements 1 and 2*). We also found that (±)-DR103 was a more potent inhibitor of AR F876L than AR WT (*Figure 4—figure supplement 3*), a phenomenon we are currently working to understand.

This striking structure–activity relationship prompted us to empirically investigate the significance of the position of the gem-dimethyl group on the D-ring of (±)-DR103. Remarkably, a compound with

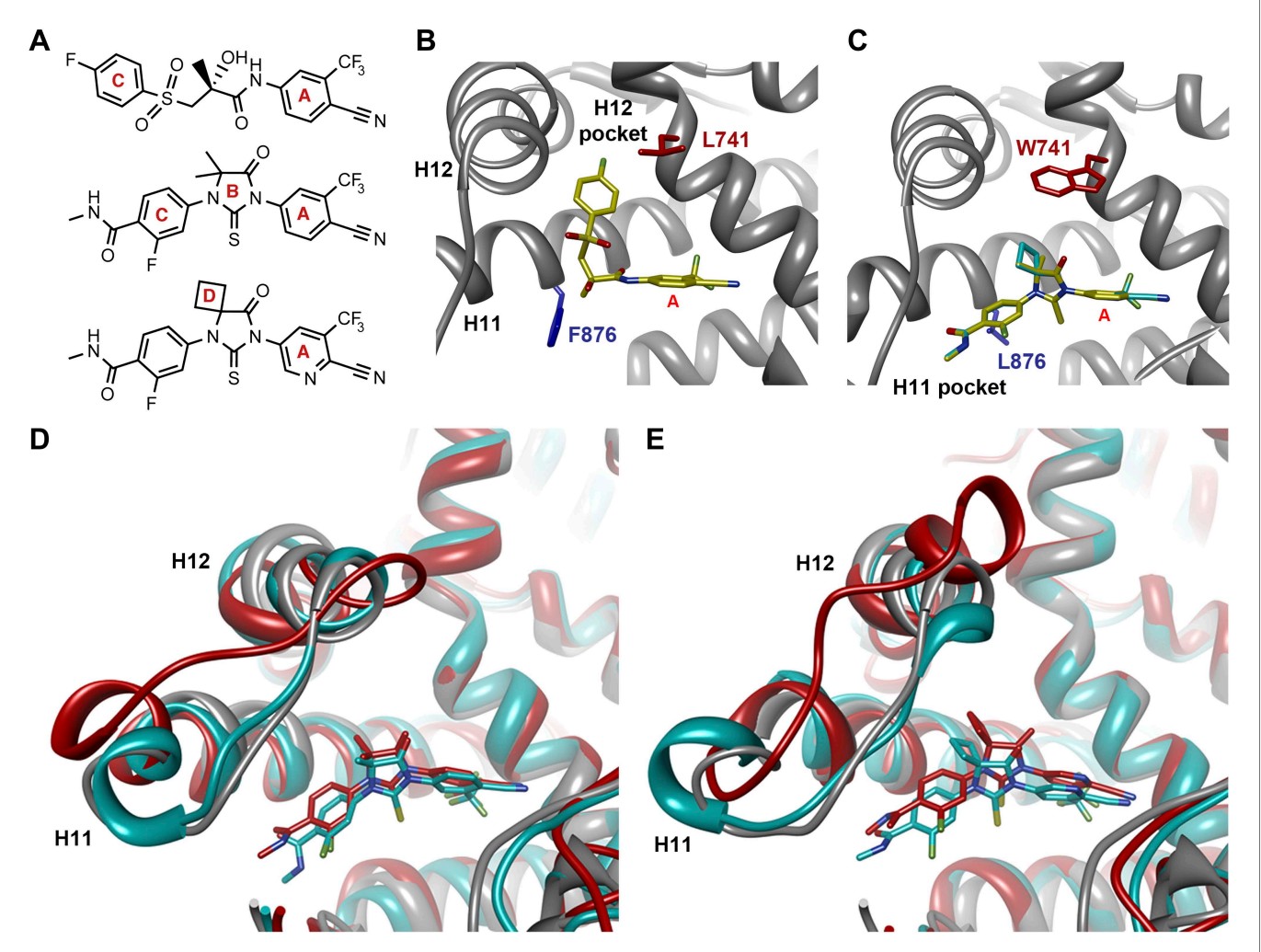

**Figure 3**. Molecular dynamics simulations predict a novel binding mode for bisaryl-thiohydantoin antiandrogens and the basis for agonism toward AR F876L. (**A**) Structures of the antiandrogens bicalutamide (top), enzalutamide (middle), and ARN-509 (bottom) oriented to highlight the common and discrete regions of the molecules. The A–D annotation of the rings is indicated. (**B**) A magnified view of the cocrystal structure of AR W741L (gray) and bicalutamide (gold) shows the antiandrogen's spatial relationship to the H11 and H12 pockets and residue F876 (blue). In this agonist conformation, the C-ring of bicalutamide does not interact with F876. (**C**) A magnified view of the initial-docked models of enzalutamide (gold) and ARN-509 (cyan) calculated using coordinates from 2AXA in which residue 741 is a tryptophan. The model suggests that the loss of torsional freedom imposed by the thiohydantoin B-ring imposes conformational restrictions on the antagonists that force the C-ring toward F876 and the 'H11 pocket'. (**D**) The lowest-energy 10-ns MD models for enzalutamide with AR WT (red) and AR F876L (cyan) overlaid on 1Z95 (gray—agonist reference structure). The F876L mutation allows for cooperative changes in neighboring residues which, when bound to enzalutamide, enable H12 to adopt a more agonist-like conformation. (**E**) An analogous view of the lowest-energy 10-ns MD models for ARN-509 with AR WT (red) and AR F876L (cyan) overlaid on 1Z95 (gray) shows a similar effect for F876L on the positioning of H11 and H12. These simulations also point to the comparatively larger dislocation in H12 by ARN-509 in AR WT, presumably owing to favorable steric interactions between the spirocyclobutyl ring and H12. AR: androgen receptor; WT: wild-type.

The following figure supplements are available for figure 3:

**Figure supplement 1**. Overlay of predicted enzalutamide or ARN-509 simulations and solved agonist crystal structure.

**Figure supplement 2**. A zoomed in view of the H11 pocket from the enzalutamide and ARN-509 MD simulations.

the gem-dimethyl group on position 4 (rather than 3/5) of the D-ring (DR104) was a modest agonist of AR F876L (albeit an antagonist of AR WT). Moreover, a compound with gem-dimethyl groups at the 3 and 5 positions of the D-ring (DR105) inhibited AR F876L (and AR WT) (*Figure 4—figure supplement 1C*), further underscoring the biological importance of the steric interactions brought about by these moieties

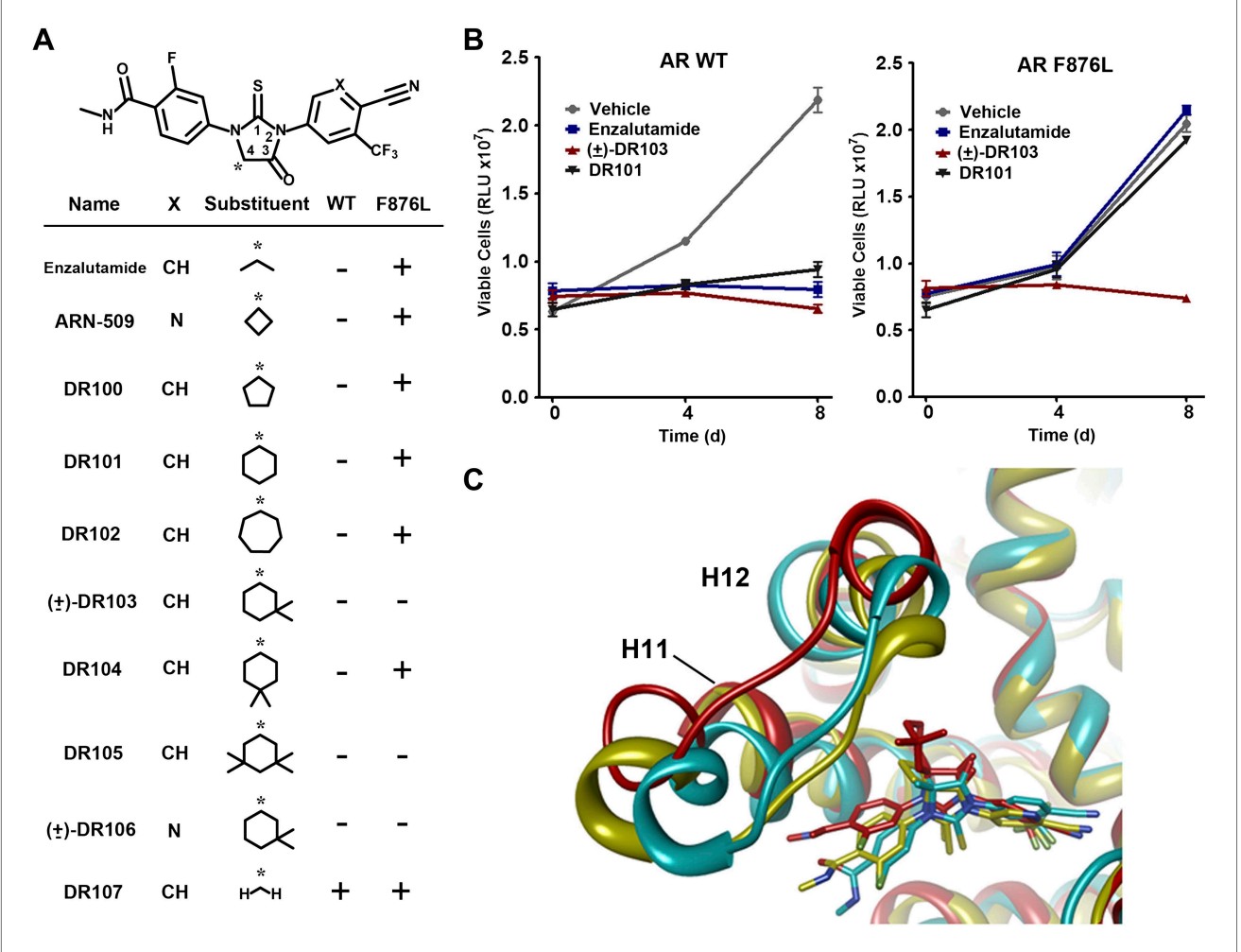

**Figure 4**. A focused chemical screen identifies novel antagonists of AR F876L. (**A**) A tabular summary of the bioactivity of the novel antiandrogens in the LNCaP/AR/Pb.PSE.EGFP cell-based assay shows the importance of a carefully designed D-ring for competent inhibition of AR F876L. Antagonism is indicated with a '−' symbol, and agonism is indicated with a '+' sign. The asterisk is situated over the shared carbon atom in position 4 that joins the bisaryl-thiohydantoin scaffold to the respective 'substituent'. The source data is outlined in *Figure 4—figure supplement 1*. (**B**) A proliferation assay for VCaP prostate cancer cells overexpressing either wild-type AR (left) or AR F876L (right) shows that (±)-DR103 effectively inhibits the growth of both models, while enzalutamide and the close structural analogue DR101 only inhibit the growth of VCaP/AR WT. Data are reported as mean ± SD, *n* = 3. (**C**) A view of the lowest-energy conformations of enzalutamide (cyan), ARN-509 (gold), and (*S*)-DR103 (red) in complex with AR F876L highlights the greater dislocation of H12 and the loop between H11 and H12 uniquely conferred by (*S*)-DR103. The color scheme invoked for AR F876L matches the respective antiandrogen. AR: androgen receptor; WT: wild-type.

The following figure supplements are available for figure 4:

**Figure supplement 1**. EGFP reporter assay for AR activity with DR series compounds.

**Figure supplement 2**. (±)-DR103 effectively competes with DHT for AR binding and induction of AR-regulated luciferase.

**Figure supplement 3**. (±)-DR103 is a more potent antagonist for AR F876L than AR WT.

**Figure supplement 4**. Growth inhibition of CWR22Pc cells overexpressing AR WT or AR F876L with (±)-DR103 treatment.

**Figure supplement 5**. Novel antiandrogen, (±)-DR103, efficiently inhibits AR signaling and induces PARP cleavage in cells expressing both AR WT and AR F876L.

*Figure 4. Continued on next page*

*Figure 4. Continued*

**Figure supplement 6**. DU145 cells treated with (±)-DR103 and DR104 display no significant growth inhibition.

**Figure supplement 7**. Overlay of predicted DR103 simulations and solved agonist crystal structure.

**Figure supplement 8**. A zoomed in view of the H11 and H12 pockets of the MD simulations for AR F876L.

in the context of the mutant receptor. Encouragingly, transplanting the D-ring from (±)-DR103 onto the ARN-509 scaffold ([±]-DR106) also resulted in AR F876L inhibition (*Figure 4—figure supplement 1B*). We interpret this result to be supportive of the model advanced by the previous MD simulations, as the F876L substitution appeared to impact the ability of enzalutamide and ARN-509 to induce H12 conformational choices in a roughly equivalent manner. Finally, to underscore the importance of the steric interactions conferred by the D-ring, we synthesized DR107, a compound built on the enzalutamide scaffold bearing only hydrogen atoms at position 4 on the B-ring. This molecule was an agonist both for AR WT and AR F876L (*Figure 4—figure supplement 1C*), pointing directly to the pharmacological significance of interactions between H12 and the substituent at the position 4 of the B-ring.

In line with this pharmacology, (±)-DR103 inhibited the growth of prostate cancer cell lines expressing both the WT and mutant receptor (*Figure 4B*, *Figure 4—figure supplement 4*). (±)-DR103 also inhibited endogenous AR signaling and induced PARP cleavage (*Figure 4—figure supplement 5*). DR101, a close structural analogue that behaved as an agonist for AR F876L, did not inhibit cell growth at equivalent doses (*Figure 4B*). Finally, the dose of (±)-DR103 required to observe antiproliferative effects (10 μM) did not impact the growth of DU145 (an AR-null human prostate cancer cell line), supporting the specificity of the antiandrogen (*Figure 4—figure supplement 6*).

Structural modeling studies for (±)-DR103 reinforced our pharmacological model for AR antagonism by bisaryl-thiohydantoins. Unlike the results for enzalutamide and ARN-509, MD simulations using (*S*)-DR103 suggested that an agonist-like conformation of H12 cannot be achieved for either WT or mutant AR (*Figure 4C*, *Figure 4—figure supplement 7*). The modeling study instead showed that the D-ring on (*S*)-DR103 was capable of directly displacing the N-terminal residues of H12. A magnified view of the H12 pocket (*Figure 4—figure supplement 8*) shows that (*S*)-DR103 occupies a region in the H12 pocket that neither enzalutamide nor ARN-509 can access, thus imposing antagonist-like dislocation of helix 12. Whereas in the H11 pocket (*Figure 4—figure supplement 8*), (*S*)-DR103 did not show a significant difference in binding with residue L876 compared to either enzalutamide or ARN-509, which suggests that the restored antagonism was not achieved by simply regaining interactions at the mutation site in AR. Similar MD simulations for the complex of AR F876L and (*R*)-DR103 showed a slightly less pronounced H12 dislocation (*Figure 4—figure supplement 7*), suggesting that the two enantiomers might cause different levels of antagonism.

## Discussion

As is evident from previous work with ABL kinase inhibitors for chronic myeloid leukemia and antivirals for HIV and hepatitis (*Shah et al., 2004*; *Glickman and Sawyers, 2012*), understanding mechanisms of drug resistance is a crucial first step in developing strategies to prevent or overcome it. With its recent approval, the case for defining mechanisms that overcome enzalutamide therapy is timely and compelling. Because AR mutations are a cause of clinical resistance to antiandrogens (flutamide and bicalutamide) (*Veldscholte et al., 1990*; *Haapala et al., 2001*), and previous work has shown that clinically relevant mutations can be discovered from screening platforms in preclinical models, we prospectively searched for such mutations in the context of enzalutamide using a novel saturation mutagenesis approach. This screen revealed that mutation of Phe 876 to Leu converts enzalutamide and ARN-509 into AR agonists and confers resistance to drug-induced growth inhibition in vitro and in vivo. Importantly, this mutation was also recovered 'spontaneously' from enzalutamide-sensitive cell line and xenograft models treated with prolonged enzalutamide therapy.

That prostate cancer can spontaneously acquire gain-of-function mutations in AR (rather than acquiring mutations that simply preclude inhibitor binding) underscores the special challenge in pharmacologically overcoming this mechanism of resistance. By borrowing insight from studies of the progesterone receptor, showing that a single amino acid can determine sensitivity to RU486, and

structural analyses of the estrogen receptor (*Benhamou et al., 1992*; *Shiau et al., 1998*), our attention was immediately directed to establishing and testing a structural model of the AR/enzalutamide complex to explain enzalutamide's curious pharmacology in the context of AR F876L. Using MD simulations, a novel binding mode for the drug was identified, which provided a compelling explanation for how antagonism is retained against the bicalutamide-resistant Trp 741 mutation. More importantly, the MD simulations argued that an altered spatial orientation of enzalutamide within the AR LBD might explain the onset of agonism, as the F876L mutation appeared to reposition the drug to eliminate steric clashes that promoted H12 dislocation in AR WT. Reassuringly, several larger compounds that the MD simulations predicted could restore H12 dislocation (the 'D-ring' series) effectively antagonized AR F876L.

Because the discovery of this mutation and its companion pharmacology provided the basis for our structural model, it is difficult to envision how the importance of the D-ring might have otherwise emerged from previous (and ongoing) chemical screening efforts. Among the ~100 bisaryl-thiohydantoins published to date, several compounds bearing structurally similar moieties to our bioactive series were essentially indistinguishable from enzalutamide, ARN-509, or other leading agents in conventional cell-based assays. Our own focused chemical screen further speaks to the unusually complicated pharmacobiology of AR F876L, as subtle changes in the position of geminal dimethyl moieties on DR103-5 radically impacted the respective bioactivity of the drugs.

Our success predicting the pharmacology of candidate inhibitors with MD simulations argues for a novel workflow by which in silico screening could guide future antiandrogen drug discovery (pending a cocrystal structure of an AR/antagonist complex). The data indicating that ~50% of patients fail to respond to enzalutamide has somewhat overshadowed the importance of the discovery of the bisaryl-thiohydantoin chemotype for AR, and the ongoing enthusiasm for developing better drugs based on this motif is most visibly reflected by the clinical trial with ARN-509. In this regard, our structural model provides a powerful tool to further refine the chemotype into drug candidates with improved properties.

Collectively, these findings demonstrate the importance of coordinated mutagenesis, structural modeling, and medicinal chemistry studies in designing drugs against an important cancer target for which appropriate drug affinity and binding conformation are mutually indispensable for competent inhibition. We are optimistic that discovery of the AR F876L mutation will facilitate solution of the enzalutamide/AR complex by X-ray crystallography. As for the potential clinical impact of a priori discovery of drug–resistance mutations to novel cancer drugs, our previous experience with the ABL kinase inhibitor dasatinib in chronic myeloid leukemia serves as an example. Within 2 years of reporting dasatinib-resistant mutations in BCR-ABL in a preclinical model, analogous mutations were recovered from dasatinib-resistant chronic myeloid leukemia patients (*Burgess et al., 2005*; *Shah et al., 2007*). We hope that this report will guide a similar search for AR mutations in prostate cancer patients who develop clinical resistance to enzalutamide. Routine rebiopsy of tumor tissue in men with castration-resistant prostate cancer is challenging due to the high frequency of osteoblastic bone lesions consisting primarily of stromal tissue. Blood-based assays for AR mutation detection may be a compelling alternative, based on recent success in detection of tumor-specific mutations in circulating plasma DNA from patients with other cancers (*Diehl et al., 2008*; *Leary et al., 2012*).

## Materials and methods

### Materials and cell lines

Fetal bovine serum (FBS) and charcoal-stripped dextran-treated fetal bovine serum (CSS) were purchased from Omega Scientific (Tarzana, CA). Bicalutamide and hydroxyflutamide (LKT Labs, St. Paul, MN), DHT (Sigma, St. Louis, MO), and R1881 (Perkin Elmer, Waltham, MA) were commercially obtained; all other ligands were synthesized at MSKCC. Serial dilutions of all drugs were made using DMSO. Antibodies used for immunoblot assays were β-actin (AC-15; Sigma) PARP (#9541; Cell Signaling Technology, Danvers, MA), FKBP5 (IHC-00289; Bethyl, Montgomery, TX), β-tubulin (D-10), and androgen receptor (N-20) (both from Santa Cruz Biotechnology, Dallas, TX). Protein lysates were prepared in M-PER protein extraction reagent (Pierce, Rockford, IL). The chromatin immunoprecipitation assay was conducted using a kit (Upstate, Billerica, MA). Nontarget and human AR siRNA pools were from the ON-TARGETplus collection (Dharmacon, Waltham, MA). LNCaP/AR cells were previously described (*Tran et al., 2009*), and CWR22Pc cells (*Dagvadorj et al., 2008*) were provided by Marja T Nevalainen (Thomas Jefferson

University, Philadelphia, PA, USA). All other cell lines were obtained from ATCC (Manassas, VA). All LNCaP and CWR22Pc derived cells were maintained in RPMI + 10% FBS. All CV1 and VCaP derived cell lines were maintained in DMEM + 10% FBS. All oligos were ordered from Operon Biotechnologies, Huntsville, AL.

For the analogue syntheses, all chemicals were acquired from Sigma-Aldrich at highest purity available and were used without further purification. Chromatography was done using Merck grade silica gel 60, and reactions were monitored by LC-MS (Waters Autopure and Acquity systems in reverse phase and with mass, evaporative light scattering, and diode array detections). Proton NMR experiments were executed on Bruker Advance DRX running at 500 MHz, and fluorine NMR was run on the same machine but at 235 MHz. Chemical shifts are reported in parts per million relative to tetramethylsilane.

## Plasmids and cell transduction

The human AR cDNA plasmid, pWZL-AR, was provided by William Hahn (Dana-Farber Cancer Institute, Boston, MA, USA). All mutant AR constructs were generated in pWZL-AR with the QuikChange II XL site-directed mutagenesis kit (Agilent, Santa Clara, CA) and primers designed using Agilent's online QuikChange Primer Design tool. Stable cell lines were generated by pantropic retroviral infection (Clontech, Mountain View, CA) and selected with blasticidin (Invivogen, San Diego, CA).

LNCaP cells were infected with the lentiviral AR-regulated EGFP reporter construct, Pb.PSE.EGFP (*Chapel-Fernandes et al., 2006*), provided by Claude Bignon (EFS Alpes Méditerranée, Marseilles, France). We then single-cell cloned the LNCaP-Pb.PSE.EGFP cells to reduce the heterogeneity in EGFP expression, and isolated a clone that had a high level of EGFP expression, which was modulated effectively by antiandrogens and AR agonists. This clone was used for all flow cytometry assays and for the FACS-based resistance screens.

## Flow cytometry analysis and FACS-sorting

LNCaP-Pb.PSE.EGFP cells for flow cytometric analysis were treated with antiandrogens (1 or 10 μM) for 4–6 days, changing media and drug every 2–3 days. Cells were collected using Accumax dissociation solution (Innovative Cell Technologies, San Diego, CA), and dead cells were counterstained using TO-PRO3-Iodide (Invitrogen, Grand Island, NY). EGFP expression was measured using the BD-FACSCalibur flow cytometer using the 488-nm laser and 530/30 bandpass filter to detect EGFP expression, and the 633-nm laser and 661/16 bandpass filter to detect TO-PRO3-Iodide labeled dead cells. For each sample, $2–5 \times 10^4$ cell events were collected and analysis was done using FlowJo software. FACS-sorting of LNCaP-Pb.PSE.EGFP cells was performed on a BD FACSVantage cell sorter. Dead cells were counterstained with DAPI (Invitrogen). EGFP expression was detected using the 488-nm laser and 530/30 bandpass filter, and DAPI-labeled dead cells were detected using the 355-nm laser and 450/50 bandpass filter.

## FACS-based bicalutamide proof-of-concept screen

We introduced four additional synonymous mutations into our pWZL-AR W741C construct to aid in distinguishing wild-type (WT) AR and AR W741C, using the QuikChange Multi Site-Directed Mutagenesis Kit (Agilent). We then designed and optimized quantitative PCR primers across these mutation sites, so that they specifically amplified AR W741C. We overexpressed AR WT or AR W741C in our LNCaP-Pb.PSE.EGFP reporter cells, mixed different ratios of cells expressing either WT or W471C, treated these cells with 1 μM bicalutamide for 4 days, and FACS-sorted the cells that maintained/induced EGFP expression. Gates for EGFP positivity were set using WT or W741C expressing cells treated with bicalutamide. Sorted cells were expanded in culture (without drug) until they reached approximately 60 million cells, we then isolated gDNA and froze down a small fraction, and the brief bicalutamide treatment and sorting was repeated on the remainder.

## FACS-based enzalutamide resistance screen

Our randomly mutagenized AR cDNA library was generated as follows: we transformed the DNA-repair-deficient *Escherichia coli* strain XL-1 Red (Agilent) with the pWZL-AR plasmid and plated them on ampicillin-agar bacterial plates. After a 36-hr incubation, colonies were collected by scraping, and plasmid DNA was purified using a plasmid MAXI kit (Qiagen, Germantown, MD). This mutagenized AR plasmid stock was used to make pantropic retrovirus (Clontech) and infect LNCaP-Pb.PSE.EGFP cells at a MOI < 1. Cells were selected for stable expression of our mutant pWZL-AR library using the blasticidin resistance cassette. Mutant library cells were cultured in 1 μM enzalutamide for 4–6 days, collected with Accumax and resuspended in Accumax containing 0.5% BSA and 10 mM HEPES. Cells

that remained EGFP positive in the presence of enzalutamide were then FACS-sorted. Gates for EGFP positivity were set using LNCaP-Pb.PSE.EGFP cells transduced with the wild-type AR cDNA, treated with vehicle or 1 μM enzalutamide. Sorted cells were expanded in culture (without drug) until they reached approximately 60 million cells, we then isolated gDNA and froze down a small fraction, and the brief enzalutamide treatment and sorting was repeated on the remainder. We performed the screen in triplicate, with five rounds of FACS and expansion for each replicate.

## AR mutation detection

Exons 2 through 8 of the exogenously expressed AR cDNA were amplified from genomic DNA isolated from cells after each sort, by high-fidelity PCR (Qiagen, Hotstar) on a Mastercycler (Eppendorf). The PCR product was subjected to bidirectional Sanger sequencing, using previously published primers (*Watson et al., 2010*). Alignments were performed using SeqMan Pro (DNASTAR), and Sanger traces were analyzed using 4Peaks software.

## qRT-PCR

Total RNA was isolated using the QiaShredder kit (Qiagen) for cell lysis and the RNeasy kit (Qiagen) for RNA purification. We used the High Capacity cDNA Reverse Transcription Kit (Applied Biosystems, Grand Island, NY) to synthesize cDNA according to the manufacturer's protocol. Quantitative PCR was done in the Realplex MasterCycler (Eppendorf) using the Power SYBR Green PCR Mastermix (Applied Biosystems). Quantitative PCR for each sample was run in triplicate and each reaction contained 1 μl of cDNA in a total volume of 20 μl. PCR quantification was done using the $2^{-\Delta\Delta Ct}$ method with normalization to GAPDH as described (Applied Biosystems). All primers were used at a final concentration of 500 nM and are listed 5' to 3': GAPDH-Forward: GAAGGTGAAGGTCGGAGTC; GAPDH-Reverse: GAAGATGGTGATGGGATTTC; PSA-Forward: GGTGACCAAGTTCATGCTGTG; PSA-Reverse: GTGTCCTTGATCCACTTCCG; Tmprss2-Forward: CACTGTGCATCACCTTGACC; Tmprss2-Reverse: ACACGCCATCACACCAGTTA; Fkbp5-Forward: TCCCTCGAATGCAACTCTCT; Fkbp5-Reverse: GCCACATCTCTGCAGTCAAA; SGK1-Forward: GCAGAAGGACAGGACAAAGC; SGK1-Reverse: CAGGCTCTTCGGTAAACTCG.

## Chromatin immunoprecipitation (ChIP)

LNCaP cells ($10^7$ cells/condition) were grown in phenol red free RPMI media supplemented with 10% CSS for 4 days, then treated with DMSO, 10 μM antiandrogens, or 1 nM DHT for 4 hr. The cells were cross-linked using 1% paraformaldehyde (Electron Microscopy Sciences, Hatfield, PA) for 15 min, glycine was then added, and samples centrifuged (4°C, 2500 rpm, 5 min) to stop further cross-linking. ChIP was performed according to manufacturer's protocols using a ChIP assay kit (Upstate) with an antibody for AR (PG-21; Upstate). Immunoprecipitated DNA was amplified by quantitative real-time PCR (ABI Power SYBR Green PCR mix). All primers were used at 500 nM and are listed 5' to 3': PSA enhancer-Forward: ATGTTCACATTAGTACACCTTGCC; PSA enhancer-Reverse: TCTCAGATCCAGGCTTGCTTACTGTC; FKBP5 enhancer-Forward: CCCCCTATTTTAATCGGAGTAC; FKBP5 enhancer-Reverse: TTTTGAAGAGCACAGAACACCT.

## Fluorescence microscopy

LNCaP cells ($10^6$ cells/well of six-well plate) were transfected with 2 μg AR-EYFP plasmid (from Jeremy Jones and Marc Diamond, UCSF) or AR.F876L-EYFP plasmid (QuikChange II XL site-directed mutagenesis kit) using FUGENE HD (Roche, Indianapolis, IN). 6 hr after transfection, media was removed and replaced with phenol red-free RPMI media supplemented with 10% CSS. The next day cells were split and plated onto poly-lysine-coated Nunc Labtek chamber slides in RPMI + 10% CSS containing DMSO, 1 μM antiandrogens, or 1 nM DHT. 24 hr later, the cells were counterstained with NucBlue Live Cell Stain Hoechst 33342 (Molecular Probes, Grand Island, NY) fixed with 4% paraformaldehyde, and mounted with a coverslip. Images were taken on a Leica TCS SP5-II Upright confocal microscope (MSKCC Microscopy Core and were analyzed for EYFP [AR] nuclear/cytoplasmic localization using ImageJ).

## AR luciferase reporter assay

CV1 cells ($2 \times 10^6$ cells/10 cm plate) were cotransfected with 50 ng of SV40 Renilla Luciferase, 5 μg of ARE(4X)-Luciferase, and 10 μg of one pWZL-AR expression construct using Lipofectamine 2000 (Invitrogen). Transfection media was removed 4–6 hr later and replaced with phenol red-free DMEM containing 10% CSS. The following day each plate was split into 48-well plates, in 10% CSS media,

containing the indicated drugs in triplicate. Luciferase activity was assayed 24–48 hr later using Dual-Luciferase Reporter Assay System (Promega, Madison, WI).

## Ligand binding assay

The binding affinity of enzalutamide to AR WT and AR F876L, relative to dihydro-testosterone (DHT), was determined using a competition assay in which increasing concentrations of cold competitor are added to cells preincubated with $^{18}$F-FDHT. LNCaP/AR WT or LNCaP/AR F876L cells were cultured in phenol red-free RPMI + 10% CSS for 2 days prior to the binding assay. Cells were trypsinized, washed in PBS, and mixed with 20,000 cpm $^{18}$F-FDHT and increasing amounts of cold competitor (10 pM–10 µM), in triplicate. The solutions were shaken on an orbital shaker at ambient temperature for 1 hr, then isolated, and washed with ice-cold tris-buffered saline using a Brandel cell harvester (Gaithersburg, MD, USA). Samples were counted using a scintillation counter, and the specific uptake of $^{18}$F-FDHT was determined. These data were plotted against the concentration of the cold competitor to give sigmoidal displacement curves, and IC$_{50}$ values were determined using a one-site model and a least squares curve fitting routine (Origin; OriginLab, Northampton, MA, USA) with the R$^2$ of the curve fit being >0.99.

## Xenograft experiments

In vivo xenograft experiments were done by subcutaneous injection of $2 \times 10^6$ LNCaP/AR cells ectopically expressing AR WT or AR F876L (100 µl in 50% Matrigel [BD Biosciences, San Jose, CA] and 50% growth media) into the flanks of castrated male SCID mice. Daily gavage treatment (using a formulation of 1% carboxymethyl cellulose, 0.1% Tween-80, 5% DMSO) was initiated on the day of injection. Once tumors were palpable, tumor size was measured weekly in three dimensions (l × w × d) with calipers. All animal experiments were performed in compliance with the guidelines of the Research Animal Resource Center of the Memorial Sloan-Kettering Cancer Center.

Xenograft experiments in which AR F876 mutations emerged after long-term treatment with second-generation antiandrogens were performed as follows: $2 \times 10^6$ LNCaP/AR cells (*Tran et al., 2009*) were injected subcutaneously into the flanks of castrated SCID mice. Treatment with 30 mg/kg enzalutamide or ARN-509 was initiated once tumors reached ~300 mm$^3$, resulting in rapid tumor regression. After several months of continual dosing, these tumors regain the ability to grow. Once these 'resistant' tumors reached their original volume, the mice were sacrificed, and tumors collected for analysis.

## CWR22Pc drug-resistant cell lines

CWR22Pc cells were cultured in RPMI + 10%FBS containing 0.1 nM DHT and either 10 µM enzalutamide or ARN-509. Treatment media was replaced every 4–5 days, and cells were passaged upon reaching confluence. Cell strains were designated as antiandrogen resistant when the time between consecutive passages was reduced to 4–6 days, which is a period of time equivalent to that of untreated CWR22Pc.

## Deep sequencing of AR

Genomic DNA (gDNA) was isolated (PureGene Core Kit A; Qiagen) from resistant CWR22Pc cell lines or LNCaP/AR xenograft tumors. With 20 ng of gDNA as template, exon 8 of AR was PCR amplified with Kapa HiFi Ready Mix (Kapa Biosystems, Woburn, MA). RNA was extracted from LNCaP/AR xenograft tumors, reverse transcribed (High Capacity cDNA Reverse Transcription Kit; Applied Biosystems), and exons 2 through 8 of AR was PCR amplified using 200 ng cDNA as template (Qiagen, HotStar).

PCR reactions were cleaned up with AMPure XP (Beckman Coulter Genomics), and pooled reaction yields were quantified using the Qubit fluorometer (Invitrogen). Library preparation was done using Nextera DNA Sample Preparation kit (Illumina) and run on the Illumina MiSeq sequencer using the 2 × 250 paired-end cycle protocol.

Genomic DNA was aligned to the hg19 build of the human genome using BWA (*Li and Durbin, 2009*) with duplicate removal using samtools (*Li et al., 2009*) as implemented by Illumina MiSeq Reporter. cDNA FASTQ files were processed with a windowed adaptive trimming tool sickle (https://github.com/najoshi/sickle) using a quality threshold of 32. The reads were then mapped to the human genome build hg19 with TopHat 2 (*Trapnell et al., 2009*) using known AR transcripts NM_000044 and NM_001011645. Duplicates were then removed with Picard (http://picard.sourceforge.net). Variant detection was performed using VarScan 2 (*Koboldt et al., 2012*) with thresholds of a minimum of 10 supporting variant reads and variant allele frequencies of at least 1%.

## Analogue syntheses

### General strategy

The syntheses were executed according to a general schema, which involves starting from a given ketone and reacting it under Strecker reaction conditions, using sodium cyanide and 4-amino-2-fluoro-N-methylbenzamide. The resulting cyanamine was then reacted with an aniline or 5-aminopyridine in the present of thiophosgene to give the desired thiohydantoins after acid hydrolysis of intermediate imine.

**General synthesis schema.**

Below are two general procedures that apply to all molecules described below.

### Strecker reaction

To a mixture of 4-amino-2-fluoro-N-methylbenzamide (0.3 mmol) and the desired ketone (1.0–2.0 equivalents [eq]) in glacial acetic acid (2 ml) was added NaCN (100 mg, 2.0 mmol, 7.0 eq), and the mixture was heated to 80°C overnight. The solvent was then removed under reduced pressure, and the residue was dissolved in water (20 ml) and then pH was brought to neutrality with aqueous saturated $NaHCO_3$ solution. Extraction with ethyl acetate (3 × 50 ml), brief drying over $Na_2SO_4$, and concentration of the filtrate under reduced pressure and the residue was chromatographed on a short path silica gel column using the gradient hexane/ethyl acetate 2/1 to 1/1.5 (vol/vol) to yield desired product in more than 85% yield.

### Thiohydantoin synthesis

Thiophosgene (5.1 µl, 66 µmol) is added dropwise to a solution of 5-amino-2-cyano-3-trifluoromethylpyridine or 4-amino-2-(trifluoromethyl)benzonitrile (60 µmol) and the given Strecker products above N-methyl-4-(1-cyanocycloalkylamino)-2-fluorobenzamides (60 µmol) in dry DMA (0.6 ml) under Argon at 0°C. After 5 min, the solution is stirred overnight at 60°C. At room temperature, this mixture was then diluted with MeOH (1 ml) and aqueous 2.0 N HCl (0.5 ml) and then the reaction was brought to reflux for 2 hr. After cooling to ambient temperature, the reaction mixture was poured into ice water (10 ml) and extracted with EtOAc (3 × 20 ml). The organic layer was briefly dried over $MgSO_4$, concentrated, and the residue chromatographed on silica gel using the gradient system hexane/ethyl acetate 2/1 to 1.5/1 (vol/vol) to yield the desired thiohydantoin in up to 90%.

**Chemical structure 1.** DR100, 4-(3-(4-cyano-3-(trifluoromethyl)phenyl)-4-oxo-2-thioxo-1,3-diazaspiro[4.4]nonan-1-yl)-2-fluoro-N-methylbenzamide.

This compound was isolated as an off-white foam.

[1]HNMR (CDCl$_3$): δ: 8.28 (t, 1 H, J = 8.5 Hz), 7.79 (d, 1 H, J = 8.3 Hz), 7.96 (bs, 1 H), 7.84 (dd, 1 H, J = 8.3 Hz, J = 1.5 Hz), 7.27 (dd, 1 H, J = 8.3 Hz, J = 1.8 Hz), 7.17 (dd, 1 H, J = 11.7 Hz, J = 1.5 Hz), 6.71 (m, 1 H), 3.07 (d, 3 H, J = 4.7 Hz), 2.36 (m, 2 H), 2.16 (m, 2 H), 1.91 (m, 2 H), 1.56 (m, 2 H).

$^{19}$FNMR (CDCl$_3$) δ: −61.98, −110.64.
LRMS for C$_{23}$H$_{18}$F$_4$N$_4$O$_2$S [M+H]$^+$ found: 491.22; calculated: 491.12

**Chemical structure 2.** DR101, 4-(3-(4-cyano-3-(trifluoromethyl)phenyl)-4-oxo-2-thioxo-1,3-diazaspiro[4.5]decan-1-yl)-2-fluoro-N-methylbenzamide.

The compound was obtained as an off-white foam.
$^1$HNMR (CDCl$_3$): δ: 8.27 (t, 1 H, J = 8.4 Hz), 7.98 (d, 1 H, J = 8.3 Hz), 7.93 (bs, 1 H), 7.82 (dd, 1 H, J = 8.2 Hz, J = 1.6 Hz), 7.19 (dd, 1 H, J = 8.3 Hz, J = 1.8 Hz), 7.08 (dd, 1 H, J = 11.6 Hz, J = 1.6 Hz), 6.70 (m, 1 H), 3.08 (d, 3 H, J = 4.7 Hz), 2.07 (m, 4 H), 1.70 (m, 6 H).
$^{19}$FNMR (CDCl$_3$) δ: −61.97, −110.92.
LRMS for C$_{24}$H$_{20}$F$_4$N$_4$O$_2$S [M+H]$^+$ found: 505.30; calculated: 505.13.

**Chemical structure 3.** DR102, 4-(3-(4-cyano-3-(trifluoromethyl)phenyl)-4-oxo-2-thioxo-1,3-diazaspiro[4.6]undecan-1-yl)-2-fluoro-N-methylbenzamide.

This compound was isolated as off-white solid.
$^1$HNMR (CDCl$_3$): δ: 8.28 (t, 1 H, J = 8.4 Hz), 7.98 (d, 1 H, J = 8.3 Hz), 7.93 (bs, 1 H), 7.82 (dd, 1 H, J = 8.2 Hz, J = 1.6 Hz), 7.24 (dd, 1 H, J = 8.3 Hz, J = 1.6 Hz), 7.14 (dd, 1 H, J = 11.6 Hz, J = 1.5 Hz), 6.72 (m, 1 H), 3.08 (d, 3 H, J = 4.7 Hz), 2.28 (m, 2 H), 2.17 (m, 2 H), 1.81 (m, 2 H), 1.60 (m, 2 H), 1.44 (m, 2 H), 1.32 (m, 2 H).
$^{19}$FNMR ( CDCl$_3$) δ: -61.98, −110.82.
LRMS for C$_{25}$H$_{22}$F$_4$N$_4$O$_2$S [M+H]$^+$ found: 519.38; calculated: 519.15.

**Chemical structure 4.** (±)-DR103, 4-(3-(4-cyano-3-(trifluoromethyl)phenyl)-7,7-dimethyl-4-oxo-2-thioxo-1,3-diazaspiro[4.5]decan-1-yl)-2-fluoro-N-methylbenzamide.

Racemic DR103 was synthesized in 70% overall yield as an off-white powder.
$^1$HNMR (CDCl$_3$): δ: 8.27 (t, 1 H, J = 8.4 Hz), 7.98 (d, 1 H, J = 8.3 Hz), 7.92 (bs, 1 H), 7.80 (dd, 1 H, J = 8.2 Hz, J = 1.7 Hz), 7.17 (dd, 1 H, J = 8.3 Hz, J = 1.7 Hz), 7.07 (dd, 1 H, J = 11.6 Hz, J = 1.6 Hz),

6.70 (m, 1 H), 3.08 (d, 3 H, J = 4.7 Hz), 2.27 (m, 1 H), 2.17 (m, 1 H), 1.93 (m, 1 H), 1.67 (m, 1 H), 1.62 (m, 1 H), 1.57 (m, 1 H), 1.52 (m, 2 H), 1.20 (s, 3 H), 0.95 (s, 3 H).

$^{19}$FNMR (CDCl$_3$) δ: -61.98, −110.89.

LRMS for C$_{26}$H$_{24}$F$_4$N$_4$O$_2$S [M+H]$^+$ found: 533.33; calculated: 533.17.

**Chemical structure 5.** DR104 4-(3-(4-cyano-3-(trifluoromethyl)phenyl)-8,8-dimethyl-4-oxo-2-thioxo-1,3-diazaspiro [4.5]decan-1-yl)-2-fluoro-N-methylbenzamide.

It was isolated as an off-white powder.

$^1$HNMR (CDCl$_3$): δ: 8.30 (t, 1 H, J = 8.4 Hz), 7.98 (d, 1 H, J = 8.3 Hz), 7.93 (bs, 1 H), 7.82 (dd, 1 H, J = 8.2 Hz, J = 1.6 Hz), 7.22 (dd, 1 H, J = 8.3 Hz, J = 1.6 Hz), 7.11 (dd, 1 H, J = 11.6 Hz, J = 1.5 Hz), 6.72 (m, 1 H), 3.08 (d, 3 H, J = 4.7 Hz), 2.04 (m, 2 H), 1.93 (m, 4 H), 1.37 (m, 2 H), 0.99 (s, 3 H), 0.73 (s, 3 H).

$^{19}$FNMR (CDCl$_3$) δ: −61.98, −110.75.

LRMS for C$_{26}$H$_{24}$F$_4$N$_4$O$_2$S [M+H]$^+$ found: 533.33; calculated: 533.17.

**Chemical structure 6.** DR105 4-(3-(4-cyano-3-(trifluoromethyl)phenyl)-7,7,9,9-tetramethyl-4-oxo-2-thioxo-1, 3-diazaspiro[4.5]decan-1-yl)-2-fluoro-N-methylbenzamide.

This compound was isolated as a beige foam.

$^1$HNMR (CDCl$_3$): δ: 8.21 (t, 1 H, J = 8.4 Hz), 7.90 (d, 1 H, J = 8.3 Hz), 7.85 (bs, 1 H), 7.73 (dd, 1 H, J = 8.2 Hz, J = 1.2 Hz), 7.12 (dd, 1 H, J = 8.3 Hz, J = 1.2 Hz), 7.02 (dd, 1 H, J = 11.6 Hz, J = 1.2 Hz), 6.64 (m, 1 H), 3.01 (d, 3 H, J = 4.7 Hz), 1.94 (d, 2 H, J = 14.4 Hz), 1.62 (d, 2 H, J = 14.4 Hz), 1.50 (s, 2 H), 1.17 (s, 6 H), 0.83 (s, 6 H).

$^{19}$FNMR ( CDCl$_3$) δ: -61.98, −110.89.

LRMS for C$_{26}$H$_{24}$F$_4$N$_4$O$_2$S [M+H]$^+$ found: 561.29; calculated: 561.20.

**Chemical structure 7.** (±)-DR106 4-(3-(6-cyano-5-(trifluoromethyl)pyridin-3-yl)-7,7-dimethyl-4-oxo-2-thioxo-1, 3-diazaspiro[4.5]decan-1-yl)-2-fluoro-N-methylbenzamide.

This compound was isolated as an off-white foam.

$^1$HNMR (CDCl$_3$): δ: 9.06 (d, 1 H, J = 1.9 Hz), 8.33 (d, 1 H, J = 1.9 Hz), 8.29 (t, 1 H, J = 8.4 Hz), 7.18 (dd, 1 H, J = 8.4 Hz, J = 1.6 Hz), 7.07 (dd, 1 H, J = 11.5 Hz, J = 1.5 Hz), 6.71 (m, 1 H), 3.08 (d, 3 H, J = 4.7 Hz), 2.30 (m, 1 H), 2.18 (m, 1 H), 1.94 (m, 1 H), 1.72 (m, 1 H), 1.63 (m, 1 H), 1.57 (m, 1 H), 1.52 (m, 2 H), 1.20 (s, 3 H), 0.94 (s, 3 H).

$^{19}$FNMR (CDCl$_3$) δ: -61.87, −110.71.

LRMS for C$_{25}$H$_{23}$F$_4$N$_5$O$_2$S [M+H]$^+$ found: 534.31; calculated: 534.16.

**Chemical structure 8.** DR107 4-(3-(4-cyano-3-(trifluoromethyl)phenyl)-4-oxo-2-thioxoimidazolidin-1-yl)-2-fluoro-N-methylbenzamide.

This compound was isolated as a white to off-white powder.

$^1$HNMR (CDCl$_3$): δ: 8.26 (t, 1 H, J = 8.4 Hz), 8.02 (d, 1 H, J = 8.3 Hz), 7.91 (bs, 1 H), 7.79 (m, 2 H), 7.45 (dd, 1 H, J = 10.7 Hz, J = 1.3 Hz), 6.71 (m, 1 H), 4.71 (s, 2 H), 3.06 (d, 3 H, J = 4.7 Hz).

$^{19}$FNMR (CDCl$_3$) δ: -62.05, −110.31.

LRMS for C$_{19}$H$_{12}$F$_4$N$_4$O$_2$S [M+H]$^+$ found: 437.19; calculated: 437.07.

## Initial models of AR–antiandrogen complex structures

No structures have been solved experimentally for enzalutamide or ARN-509 in complex with AR (agonist or antagonist conformation). Therefore, three-dimensional structures of antiandrogens were first built using the computer program Gaussview (version 4.1.2; part of the computer program Gaussian 03) (*Frisch et al., 2004*) and then geometrically optimized in a quantum mechanical force field at the level of restricted Hartree-Fock (RHF) 6-31g* using the program Gaussian 03. The partial atomic charges were derived from the optimized structures by Restrained ElectroStatic Potential (*Bayly et al., 1993*; *Cornell et al., 1993*) (RESP) fitting to the RHF/6-31g* potentials. The other parameters modeling the antiandrogens were taken from the CHARMm22 (*Momany and Rone, 1992*) force field after assigning CHARMm22 atom types to antiandrogens with an in-house program.

The initial AR–antiandrogen complex structures were then modeled with the molecular modeling program CHARMM (*Brooks et al., 1983*, *2009*). Starting with the atomic coordinates of AR WT and the A-ring of S1 in the template crystal structure (PDB accession code, 2AXA), the side chain of residue F876 was replaced with CHARMm22-parameterized side chain of a leucine when needed and the CH group on the A-ring was replaced with a nitrogen in cases of ARN-509. The rest of each antiandrogen was 'grown' from the A-ring using the ideal unbound structures solved by geometry optimization. Missing side chain atoms were built using standard CHARMm22 parameters and hydrogen atoms were added with the HBUILD (*Brunger and Karplus, 1998*) module of CHARMM. All these newly introduced atoms without three-dimensional crystal coordinates treated flexible and the rest under harmonic constraints with the force constant of 100 kcal/mol/Å$^2$, each AR–antiandrogen complex structure was energetically minimized with one round of 100-step steepest decent followed by two rounds of 100-step Adopted-Basis Newton–Raphson (ABNR) energy minimization. Harmonic constraints were reset at the beginning of each round of minimization. No nonbonded cutoff was used. Solvent effects were implicitly modeled in this stage with a distance-dependent dielectric constant.

## Molecular dynamics simulations

The all-atom MD simulations were performed with explicit solvent atoms using the program CHARMM (version 36a1). Each initial AR–antiandrogen model was first centered and overlaid with a rhombic dodecahedron-shaped water box (edge length being 88 Å) of approximately 47,000

equilibrated water molecules. Any water molecule whose oxygen atom was within 2.8 Å away from any non-hydrogen atom of AR or antiandrogen was removed. Proper amount of sodium and chloride ions were automatically added to achieve overall charge neutrality and physiological level of ion concentration (0.145 M). Their positions were optimized with 10 independent trajectories of randomly replacing water molecules and performing 50 steps of steepest decent and 125 steps of ABNR energy minimization.

The molecular system including AR, antiandrogen, water, and ions was heated to 300 K and equilibrated with two rounds of 0.1-ns MD simulations under successively weaker harmonic constraints on AR or antiandrogen atoms. After the MD equilibration, three sets of random velocities were assigned to initiate three independent 10-ns MD productions. The MD equilibration and production were performed using the crystal form of rhombic dodecahedron (RHDO) and the canonical ensemble (NVT). A nonbonded cutoff of 14 Å, periodic boundary conditions in conjunction with Ewald summation method, the leapfrog Verlet integrator, and the Hoover thermostat for pressure and temperature were used. The timestep was set at 2 fs. Parallel jobs for MD simulations were run on a computer cluster of Intel Xeon X5650 series (2.66 GHz and 4 GB memory per CPU).

### Molecular visualization
Structural models were visualized in a molecular graphics program, UCSF Chimera (*Pettersen et al., 2004*). The default option used when aligning structures.

## Acknowledgements

We thank the flow cytometry core facility at MSKCC for technical support, O Ouerfelli and G Yang of the organic synthesis core facility at MSKCC for synthesizing and characterizing DR100-107, the Geoffrey Beene Translational Oncology Core at MSKCC for sequencing data, Bruce Tidor for kindly providing access to a computer cluster that allowed for faster simulations, Rohit Bose, Brett Carver and Phil Iaquinta for helpful discussions, and Anson Ku, Taslima Ishmael, Emily Schenkein, Bradley Green, Victor Cruz, Muriel Lainé, Shyamala Rajan, and KC Vavra for technical support.

## Additional information

### Competing interests
MDB, Co-inventor of patent applications covering the AR mutation and novel chemical matter described in this manuscript. MJE, DJH, GLG, YS, Co-inventor of a patent application covering the novel chemical matter described in this manuscript. PAW, Co-inventor of a patent application covering the AR mutation described in this manuscript. CLS, Senior editor, *eLife*; co-inventor of enzalutamide and ARN-509; co-founder of Aragon Pharmaceuticals; co-inventor of patent applications covering the AR mutation and novel chemical matter described in this manuscript. The other authors declare that no competing interests exist.

### Funding

| Funder | Grant reference number | Author |
| --- | --- | --- |
| National Cancer Institute | CA155169 | Charles L Sawyers |
| National Institutes of Health | R25-CA096945 | Michael J Evans |
| National Cancer Institute | CA089489 | Geoffrey L Greene |
| Virginia and D. K. Ludwig Fund | | David J Hosfield, Geoffrey L Greene |
| Geoffrey Beene Cancer Research Center | | Michael J Evans, Charles L Sawyers |
| MSKCC Experimental Therapeutics Center | | Michael J Evans, Charles L Sawyers |
| MSKCC Imaging and Radiation Sciences Bridge Program | | Michael J Evans |
| Toyota Technological Institute at Chicago | | Yang Shen |

| Funder | Grant reference number | Author |
| --- | --- | --- |
| Howard Hughes Medical Institute | | Charles L Sawyers |
| CDMRP Physician Research Training Award PC102106 | | Vivek K Arora |

The funders had no role in study design, data collection and interpretation, or the decision to submit the work for publication.

### Author contributions

MDB, MJE, YS, Conception and design, Acquisition of data, Analysis and interpretation of data, Drafting or revising the article; DJH, GLG, CLS, Conception and design, Analysis and interpretation of data, Drafting or revising the article; JW, VKA, PAW, Conception and design, Acquisition of data, Analysis and interpretation of data; YC, Conception and design of the saturation mutagenesis screen

### Ethics

Animal experimentation: This study was performed in strict accordance with the recommendations in the Guide for the Care and Use of Laboratory Animals of the National Institutes of Health. All animal experiments were conducted in compliance with Institutional Animal Care and Use Committee (IACUC) guidelines at Memorial Sloan-Kettering Cancer Center under the approved protocol #06-07-012 and institutional guidelines for the proper, humane use of animals in research were followed.

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
