## [Decision Letter]

Thank you for choosing to send your work entitled “Overcoming mutation-based resistance to antiandrogens with rational drug design” for consideration at *eLife*. Your article has been favorably evaluated by a Senior editor and 3 reviewers, one of whom is a member of our Board of Reviewing Editors, and one of whom, John Katzenellenbogen, wants to reveal his identity.

The Reviewing editor and the other reviewers discussed their comments before we reached this decision, and the Reviewing editor has assembled the following comments to help you prepare a revised submission.

This manuscript describes the structural and molecular mechanism of AR resistance to enzalutamide, a second generation of anti-androgen used for treating castration resistant prostate cancer. The authors discovered that AR mutation in F876 results in converting enzalutamide from an AR antagonist to an AR activator. Using structural modeling and simulation, the authors design a series of compounds, some of which show inhibitory activity against the mutated form of AR, and could have potential as new anti-androgen drugs.

All three reviewers found the manuscript to be of high interest and importance. One reviewer stated, “Overall, the data is convincing and comprehensive, and is of high quality. The results presented here are important and have clinical implications as enzalutamide has been recently approved by FDA for treating hormone resistant prostate cancer.” A second reviewer stated, “This is a study of remarkable interest and impact, both from a conceptual, mechanistic, and potentially clinical point of view. It is well conceived, well organized, well presented, and well documented.”

1) From a biological standpoint, the only significant experiment considered to be missing was a test of one of the novel antagonists for suppression of tumorigenicity of the F867L prostate cancer cell lines in the xenograft model. This experiment would complete the thought that the rational drug design process ends with the predicted anti-tumor effect in vivo. After deliberation, this experiment was considered not necessary for publication given the overall depth of the studies, but it would further strengthen the manuscript if available.

2) All three reviewers thought that despite the nice rendering of AR-ligand structures after MD simulations, it is not clear exactly in what way the F876L change results in altered interaction with the ENZ and ARN ligands, and where on the ligand the contact is occurring. A figure illustrating an enlargement of this region would add to the paper and perhaps enable a cogent discussion of the specific contacts the enables antagonism to is restored by altering the ligand structure.

3) The paper would be strengthened by evidence that the F876L mutation does indeed occur in PCa patients undergoing ENZ therapy, and, although it may be too soon for ENZ-resistant patients to have emerged, it would be worth discussing this as a future direction, because these patients would obviously be candidates for treatment with the second-generation drug reported in this paper.

4) In Figure 3, the distance of residues 876 to the bound enzalutamide need to be carefully measured and illustrated in both WT and mutated receptors. The implication of this distance needs to be discussed within the contact of receptor-ligand interactions.

5) Again in Figure 4, panel A, the distance of residues 876 to the bound compounds needs to be carefully measured and illustrated in both WT and mutated receptors. The implication of this distance needs to be discussed within the context of receptor-ligand interactions.

6) Results section: there is often interplay between the degree of antagonism and affinity/potency, with introduction of larger substituents engendering greater antagonism, but in most cases also lowering potency. It looks like the two new antagonists are equipotent with their ENZ and ARN parents, but this should be stated more clearly. Also, it would be reassuring and potentially interesting to have binding affinity measurements of the parent compounds, the new antagonists on both WT and F876L AR.

7) In considering the value of this work as a framework for future studies, it would be useful to know what other AR mutations, if any, were identified in the drug resistant tumors listed in Table 1. Why not just sequence drug resistant tumors rather than go through the mutagenesis step. If other mutations were identified, are they also sensitive to the novel antagonists? Some additional consideration of this point is needed.

8) Bicalutamide appears to be an antagonist on F876L AR in some Figures (such as Figure 2B) but an agonist in others. This should be commented upon.

Additional comment

9) It would be of interest to separate the enantiomers of the two new antagonists, DR103 and 106. They likely have different potencies and efficacies, and if one is more potent and more agonistic than the other, it might limit the activity measured for the racemic. The authors may wish to comment on this.

---

## [Author Response]

*1) From a biological standpoint, the only significant experiment considered to be missing was a test of one of the novel antagonists for suppression of tumorigenicity of the F867L prostate cancer cell lines in the xenograft model. This experiment would complete the thought that the rational drug design process ends with the predicted anti-tumor effect in vivo. After deliberation, this experiment was considered not necessary for publication given the overall depth of the studies, but it would further strengthen the manuscript if available*.

We agree that in vivo tumor xenograft studies would provide further strength to our rational drug design and we are embarking on pharmacokinetic and pharmacodynamic studies before we proceed with xenograft studies. These are lengthy experiments that will be reported later.

*2) All three reviewers thought that despite the nice rendering of AR-ligand structures after MD simulations, it is not clear exactly in what way the F876L change results in altered interaction with the ENZ and ARN ligands, and where on the ligand the contact is occurring. A figure illustrating an enlargement of this region would add to the paper and perhaps enable a cogent discussion of the specific contacts the enables antagonism to is restored by altering the ligand structure*.

Thank you for these suggestions. We have included additional figure panels (new figures: Figure 3–figure supplement 2, and Figure 4–figure supplement 8) and discussion of specific AR-ligand contacts in the results to further clarify the possible mechanisms underlying the antagonism-to-agonism switch for ENZ and ARN, and the restored antagonism for newly developed ligands.

*3) The paper would be strengthened by evidence that the F876L mutation does indeed occur in PCa patients undergoing ENZ therapy, and, although it may be too soon for ENZ-resistant patients to have emerged, it would be worth discussing this as a future direction, because these patients would obviously be candidates for treatment with the second generation drug reported in this paper*.

We agree that our paper would be strengthened by clinical evidence of the mutation and we are working hard to obtain relevant clinical material. This is challenging for two reasons:

First, enzalutamide was FDA approved only five months ago, so very few patients have been treated outside the original clinical trials, and, for the most part, trial investigators did not obtain tumor tissue or plasma.

Second, CRPC presents unique clinical challenges in obtaining tumor tissue that have led us to focus on developing blood-based assays to detect AR mutations (CTCs and ctDNA). We are actively pursuing these studies with collaborators.

We have updated the text to discuss this future direction.

*4) In Figure 3, the distance of residues 876 to the bound enzalutamide need to be carefully measured and illustrated in both WT and mutated receptors. The implication of this distance needs to be discussed within the contact of receptor-ligand interactions*.

We have included the measurements requested (new figure: Figure 3–figure supplement 2), and we have added discussion of these data within the text.

We have indicated the distances between residues 876 on AR WT or AR F876L to the bound enzalutamide and ARN-509. The C rings are in close proximity (less than 4 Å) to position 876. Moreover, F876 (WT) was slightly closer to the C rings than L876, and clearly formed more pairwise atomic contacts with the C rings. We have added to the Discussion as described above. Meanwhile, we avoid over-interpreting the exact values of these distances, as positions of C rings in simulated structures were seen to vary even though they remained approximately at the same spot near residues 876.

*5) Again in Figure 4, panel A, the distance of residues 876 to the bound compounds needs to be carefully measured and illustrated in both WT and mutated receptors. The implication of this distance needs to be discussed within the context of receptor-ligand interactions. We have included the measurements requested (new figure: Figure 4–figure supplement 8), and we have added discussion of these data within the text*.

We have indicated the distances between residues 876 on AR WT or AR F876L to the bound DR103. We emphasize here that the restored antagonism was not due to recovering receptor–ligand interactions at the mutation site (residue 876 on helix 11) but rather pushing helix 12 to adopt an antagonist-like conformation as our design rationale, the D-ring hypothesis, intended. The newly added inset also indicated that there is no proof of receptor–ligand interactions at the mutation site being recovered, which might be the only implication these distances provided. For the reasons above, we chose to not discuss these distances further.

*6) Results section: there is often interplay between the degree of antagonism and affinity/potency, with introduction of larger substituents engendering greater antagonism, but in most cases also lowering potency. It looks like the two new antagonists are equipotent with their ENZ and ARN parents, but this should be stated more clearly. Also, it would be reassuring and potentially interesting to have binding affinity measurements of the parent compounds, the new antagonists on both WT and F876L AR*.

This is an important point that we have investigated with additional experiments. Under conditions of relatively low androgen concentrations (10% FBS), DR103 is equipotent to enzalutamide and ARN-509. However, as predicted by the reviewers, DR103 is less potent against WT AR under conditions of high androgen concentration as measured by reporter assays (new figure: Figure 4–figure supplement 3). As for binding affinity, we have included a competitive binding experiment with radiolabeled DHT in cells expressing the WT or mutant receptor, showing increased affinity of enzalutamide for F876L (comparable to DHT affinity), consistent with conversion to agonism (new figure: Figure 2– figure supplement 3).

*7) In considering the value of this work as a framework for future studies, it would be useful to know what other AR mutations, if any, were identified in the drug resistant tumors listed in Table 1. Why not just sequence drug resistant tumors rather than go through the mutagenesis step. If other mutations were identified, are they also sensitive to the novel antagonists? Some additional consideration of this point is needed*.

We chose the mutagenesis screen strategy based on our prior success with kinase inhibitor mutagenesis screens (imatinib, dasatinib) and the speed with which mutations can be recovered relative to waiting for outgrowth of drug-resistant xenografts. We also believe this could be a general methodology for other NHRs.

Although we cannot say that we screened to saturation, F876L is the only mutation we recovered and we recovered it multiple times. A few additional AR mutations emerged in the drug resistant xenograft tumors, but F876L was one of only 2 mutations that occurred in multiple tumors, and the only mutation that arose in both the LNCaP/AR xenograft and CWR22Pc systems. The second mutation (isolated multiple times in the xenograft model) has been reported in association with androgen insensitivity syndrome (AIS) and is therefore predicted to yield a functionally impaired AR. We are in the process of characterizing this one further.

*8) Bicalutamide appears to be an antagonist on F876L AR in some Figures (such as Figure 2B) but an agonist in others. This should be commented upon*.

As noted by the reviewers, the pharmacology of bicalutamide against the mutant varies depending on the context. For example, bicalutamide is clearly not an agonist for F876L under the androgen-deprived conditions in which enzalutamide is. However, in the presence of androgen, bicalutamide at low concentrations has modest antagonist activity (as measured by our GFP reporter) that is lost at higher concentrations, presumably due to the well-documented agonism of bicalutamide against WT AR when expressed at high levels. Finally, bicalutamide has minimal antagonist activity against mutant-expressing cells in growth assays. These data are now added in a new figure (Figure 2–figure supplement 8) and in a new paragraph in the Results. Our conclusion from all of this is that bicalutamide is highly unlikely to provide a clinical solution to the F876L mutant; hence, the rationale for DR103.

*Additional comment*:

*9) It would be of interest to separate the enantiomers of the two new antagonists, DR103 and 106. They likely have different potencies and efficacies, and if one is more potent and more agonistic than the other, it might limit the activity measured for the racemic. The authors may wish to comment on this*.

This is an important point with clear implications for future drug design, as suggested by the modeling reported in this paper. While work has begun to address this issue by chromatographically separating the enantiomers, we are not in a position to include pharmacological data associated with the enantiomers until their absolute conformation has been defined. We are actively working to crystallize the compounds to define the (S) and (R) enantiomers.